# Examining the effectiveness of Prostatic hyperplasia education on the level of participant's knowledge and awareness

Hari Krismanuel[1]*, Purnamawati Tjhin[2]

1 Department of Surgery, Faculty of Medicine, Universitas Trisakti, Jakarta, Indonesia, 2 Department of Anatomy, Faculty of Medicine, Universitas Trisakti, Jakarta, Indonesia

* hari_krismanuel@trisakti.ac.id

## Abstract

Prostatic Hyperplasia (PH) is a common condition among older men, contributing significantly to Lower Urinary Tract Symptoms (LUTS). Despite available medical treatments, there is a lack of public awareness about PH, resulting in late diagnoses. This study offers a novel approach by using **a community-based educational intervention** to increase knowledge of PH, leveraging the **International Prostatic Symptom Score (I-PSS)** as a tool for educating elderly men in Bogor, Indonesia. This study aims to evaluate the effectiveness of community-based education on prostatic hyperplasia (PH) in enhancing knowledge and awareness among elderly male participants, addressing the gap in public awareness and the need for early detection, unlike prior studies conducted in clinical settings. By utilizing the International Prostatic Symptom Score (I-PSS), this study provides a structured approach to community health education and symptom self-assessment. A mixed-methods, quasi-experimental pretest-posttest design involved 32 participants aged $\geq 60$ years in Bogor, Indonesia. Quantitative data assessed changes in knowledge before and after the intervention, while qualitative insights were gathered through post-education discussions. Statistical analyses, including paired t-tests and effect size (Cohen's d), were conducted to measure the intervention's impact. This study is unique in its application of **I-PSS** for community education and its combination of quantitative and qualitative data to measure knowledge improvement and explore participant perceptions. Significant knowledge improvement was found post-intervention (mean increase: 8.9, $p < 0.001$, Cohen's $d = 0.82$). The integration of qualitative feedback highlighted the relevance and clarity of the intervention while also identifying remaining knowledge gaps, revealing its holistic impact on the participants. The novelty of this study lies in its community-based approach using I-PSS, which is an underexplored method in educating about PH. The results provide strong evidence for using structured community education to promote early detection and improve PH awareness. Future studies could benefit from including control groups and testing this approach in other

**Data availability statement:** All relevant data are available in the Supplementary Materials and have been deposited in the Figshare repository, accessible via https://doi.org/10.6084/m9.figshare.28595594. These Supplementary Materials can be accessed from the manuscript. Additionally, the Supplementary Materials have been uploaded to PLOS ONE along with the revised manuscript. The Supplemental Materials include: Table S1: A brief summary of the management options discussed during the educational session Table S2: Anonymized participant data Table S3: The detailed SPSS output of our statistical analysis Appendix S1: The complete pre-test and post-test questionnaire We confirm that all participant data presented in Table S2 have been fully anonymized to ensure confidentiality and comply with ethical research standards.

**Funding:** The author(s) received no specific funding for this work.

**Competing interests:** The authors have declared that no competing interest exist.

regions for broader applicability. This study demonstrates the effectiveness of a community-based educational intervention using the International Prostatic Symptom Score (I-PSS) in improving knowledge and awareness about Prostatic Hyperplasia (PH) among elderly men. The findings suggest that integrating tools like the I-PSS in community health programs can empower individuals to self-assess their symptoms, promote early detection, and reduce the burden of delayed diagnoses. These results underscore the potential of scalable, low-cost interventions to address health education gaps in similar low-resource settings globally.

## Introduction

Prostatic Hyperplasia (PH), also known as Benign Prostatic Hyperplasia (BPH), is a common condition among aging men, particularly those aged 50 and above. It leads to Lower Urinary TractSymptoms (LUTS), such as frequent urination, nocturia, and a weakurinary stream, significantly affecting quality of life. Despite its high prevalence, public awareness of PH remains insufficient, leading to delayed diagnoses and increased healthcare burdens. Early detection and timely management can prevent disease progression and reduce the need for invasive treatments such as surgery [1,2].

Pharmacological treatments, including alpha-blockers and 5-alpha-reductase inhi - bitors, are available but often underutilized due to insufficient public knowledge regarding the symptoms and treatment options for PH. Additionally, studies indicate that many men are hesitant to seek medical advice due to social stigma and lack of awareness, which often leads to late-stage diagnoses. Public education plays a crucial role in addressing these gaps, particularly among older populations [3,4]. Despite the availability of medical interventions, there is a lack of global initiatives focused on community-level education to address the burden of PH. Early detection and management through educational efforts can reduce complications, prevent disease progression, and lower healthcare costs. This study addresses a critical gap in existing research by introducing a community-based educational intervention designed specifically for elderly individuals diagnosed with prostatic hyperplasia, aiming to offer a scalable and accessible model for improving prostate health awareness. By leveraging the International Prostatic Symptom Score (I-PSS) as a novel tool in public health education, this study offers a unique approach that empowers individuals to self-assess their symptoms and promotes early detection in a non-clinical, community setting. Given the global increase in life expectancy and the corresponding rise in prostate health challenges, this research contributes to scalable solutions that can be adapted to diverse healthcare systems worldwide. Unlike traditional clinical approaches, using I-PSS in a community context provides broader access for participants, enabling them to gain a deeper understanding of their condition independently [5–8].

**Community - based health education interventions** have shown promise in improving public health outcomes by increasing awareness of preventable diseases.

However, limited research has focused on using educational programs specifically designed for PH awareness in low-resource settings. This study fills this gap by integrating I-PSS forms as part of a structured educational program that combines a variety of learning materials and interactive methods. These methods not only address knowledge gaps but also focus on promoting **behavioral changes** in a public setting. This holistic approach enhances the **effectiveness of prostate health education**, making it more accessible and impactful for affected individuals, and ultimately improving their quality of life [5–8].

Prior studies on PH education have predominantly been conducted in **clinical settings**, leaving a significant gap in community-based interventions. This study fills this gap by integrating I-PSS forms as part of a structured educational program that combines a variety of learning materials and interactive methods. These methods not only address knowledge gaps but also focus on promoting **behavioral changes** in a public setting. This holistic approach enhances the **effectiveness of prostate health education**, making it more accessible and impactful for affected individuals, and ultimately improving their quality of life [8–11].

This study aims to determine whether **a structured community-based educational intervention** using the International Prostatic Symptom Score (I-PSS) can effecti - vely improve knowledge and awareness about PH among elderly men. By employing a mixed-methods approach, we measure changes in knowledge and explore participant perceptions to assess the intervention's impact [8–11].

The novelty of this research lies in its integration of the I-PSS into a **community health education program**, targeting elderly men who may have limited access to formal healthcare resources. The intervention's effectiveness in raising awareness and improving knowledge is assessed through a **mixed-methods approach**, combining quantitative and qualitative data to comprehensively evaluate the program's impact. The findings of this study have the potential to inform public health strategies aimed at improving PH awareness and early detection, not only in Indonesia but also in other similar settings worldwide.

## Methods

### Research design

This study utilized **a prospective mixed methods design,** a novel combination of quantitative and qualitative methods, including pre-test/post-test measurements and interactive question-and-answer sessions to evaluate the impact of health education on participant knowledge. It was conducted in a community setting in Bogor, Indonesia on February 12, 2024 [12–14].

**Quantitative component:** This research is experimental, using a quasi-experi - mental study with one group pretest-posttest design [15,16]. This quasi-experimental design was selected due to its feasibility and ethical consider- ations in a real-world community setting. While the absence of a control group is a limitation, it allowed us to reach a larger, diverse population and provided insights into natural variations in knowledge gains. Future studies with control groups are recommended to validate and generalize these findings. **Cohen's d** was calculated to determine the effect size of the intervention on knowledge improvement. The quantitative component involved pre- and post-intervention assess- ments using structured questionnaires to measure changes in knowledge levels among participants [15–19].

In this research design, a pretest was carried out first, to determine the participants' level of knowledge and aware- ness of the symptoms and signs of prostatic hyperplasia before education. Next, information about the warning signs and symptoms of PH is given. During this instruction, learners learned about the International Prostatic Symptom Score, or I-PSS. Participants received structured instruction on all aspects of PH, with a specific focus on interpreting their I-PSS results independently. This innovative approach not only enhanced participants' understanding of their health conditions but also equipped them with actionable insights to seek timely medical attention. This strategy differs from prior studies by integrating an educational tool typically reserved for clinical evaluation into a broader public health framework. After

completing the education, an interactive question and answer session was held. The participants were allowed to ask everything about PH. After that, a post test was carried out with the same questions as the pre test questions to compare the level of knowledge and awareness of the participants before and after the education. The study was conducted in Nagrak village, Bogor regency in February 2024 [8,15].

**Qualitative Component**: The study incorporated a qualitative element through post-education question-and-answer session to enrich and contextualize the quantitative findings. This session were designed to capture participants' perceptions, experiences, and any immediate feedback regarding the health education they received. The qualitative data were transcribed and analyzed using thematic analysis to identify key themes that could inform the interpretation of quantitative results and provide deeper insight into the participants' understanding [12–14].

### Population and sample

The population in this study were men aged ≥ 60 years in Ciangsana and Nagrak villages, Gunung Putri District, Bogor Regency. This research was conducted on February 12, 2024. Here's a general formula for sample size estimation in the context of a t-test (assuming a two-tailed test) [20,21]:

$$n = \frac{2 \cdot (Z\alpha/2 + Z\beta)^2 \, SD^2}{\text{Effect Size}^2}$$

Where:
n is the required sample size.
Zα/2 is the critical value for the desired level of significance (α).
  Zβ is the critical value for the desired power (1 − β).
  SD is the estimated standard deviation in the population.
  Effect Size is the expected size of the difference between groups.
  For a **95% confidence level**, Zα/2 is **1.96**, and for **90% power**, **σ2 = 25, δ = 2,** samples size should be collected are 29.54 ≈ 30 [14–17].

### Inclusion and exclusion criteria

**Inclusion criteria** for this study included male participants aged ≥ 60 years, and residing in Nagrak Village or the surrounding area. Inclusion criteria required participants to be able to attend the session, literate and fluent in Bahasa Indonesia, able to attend the session and provide informed consent. Participants must be willing to take part in the entire series of interventions and be able to communicate and read well. **Exclusion criteria** included participants with severe comorbidities such as diabetes or heart disease, cognitive or psychological disorders, and unwillingness to participate fully, participants who could not speak Bahasa Indonesia and illiterate. In addition, participants who had received similar education previously were also excluded to avoid bias in assessing the impact of the intervention.

### Participants

After inclusion and exclusion criteria, a total of **32 elderly men** aged 60 years or older participated in this study. Participants were recruited through community outreach programs and local healthcare facilities. The participants were randomly assigned to the intervention group without a control group due to logistical and ethical considerations in the community setting.

The demographic characteristics of the study participants are summarized in Table 1. All participants had an elementary school education, ensuring uniformity in educational background and minimizing potential confounding effects related to knowledge diffe- rences.

**Table 1. Demographic Characteristics of Participants.**

| Variable | Category | n (%) |
|---|---|---|
| Age Group | 60-65 | 13 (40.62) |
| | 66-70 | 9 (28.13) |
| | >70 | 10 (31.25) |
| Education Level | Elementary School | 32 (100) |

## Educational intervention

Educational interventions focus on providing information about prostatic hyperplasia, its symptoms, available treatments, and lifestyle modifications. The format is that participants receive an education/counseling session conducted by a surgeon. The education/ counseling session lasts approximately 30 minutes. Educational materials in the form of power point files, posters, I-PSS forms and visual aids, are used to improve understanding.

The intervention consisted of a **single, structured educational session** delivered to participants after they completed the pretest. The program included a **combination of face-to-face lectures, printed materials, and interactive group discussions** focusing on the causes, symptoms, and management options for PH, with particular emphasis on the importance of early detection and lifestyle modifications. The **International Prostatic Symptom Score (I-PSS)** was introduced as a self-assessment tool for the participants to track their symptoms, empowering them to evaluate their condition independently and seek timely medical advice.

The intervention took place in a community hall to ensure accessibility for all participants. Educational materials were provided in **Bahasa Indonesia**, and entire the session lasted approximately **60 minutes,** comprising a 30-minute structured lecture followed by a 30-minute interactive group discussion. During the session, trained community health workers facilitated the discussion, answering questions and encouraging participants to complete the I-PSS form as part of the educational process.

A brief overview of management options for prostatic hyperplasia was provided during the educational session but was not included in the pretest/posttest assessment [22,23]. Further details on management options are available in Table S1 (Supplementary Materials). The dataset has been deposited in Figshare and can be accessed through this link.

## Pretest and posttest procedures

A pre-test was conducted before the educational intervention to assess participants' baseline knowledge and awareness. Participants completed the pretest individually in a supervised setting. After the educational intervention, a single interactive question-and-answer session was held to ensure participants' understanding and to address any remaining questions. After completing the interactive question-and-answer session, a post-test was administered to evaluate the impact of the intervention. The same questionnaire was used for both the pre-test and post-test to ensure consistency in measurement. The questionnaire consisted of 20 true-or-false questions designed to assess participants' understanding of lower urinary tract symptoms (LUTS) associated with Prostatic hyperplasia (PH). These questions were structured based on the International Prostatic Symptom Score (I-PSS), covering aspects such as incomplete emptying, frequency, intermittency, urgency, weak stream, straining, and nocturia. Given that participants were from a low-education background, the test format was simplified to a True (B)/ False (S) system to facilitate comprehension and response accuracy.

During the educational sessions, participants were introduced to the International Prostatic Symptom Score (I-PSS) to enhance their understanding of LUTS and its assessment. This approach ensured that participants were familiar with the symptoms assessed in both tests and minimized potential bias in self-reported responses. By providing clear explanations of PH warning signs, the intervention aimed to address limitations related to measurement tools. A one-group pretest-posttest design was employed to measure the effectiveness of the intervention [12–14].

The pre-test and post-test questions used to assess participants' knowledge before and after the community-based education session are provided in Appendix S1 (Supplementary Materials). The dataset has been deposited in Figshare and can be accessed through this link.

## Data collection

Data collection was conducted on February 12, 2024, involving 32 elderly menaged 60 years or older recruited through local community outreach programs. Data were collected through a combination of **pre- and post-intervention surveys** to assess changes in participants' knowledge and awareness. The pretest was conducted at the beginning of the program, while the posttest was administered at the conclusion of the educational session.

**Quantitative data.** Knowledge of PH was measured using a **standardized questionnaire** that assessed participants' understanding of the symptoms, risk factors, and management strategies for PH. Participants' **I-PSS scores** were also used to evaluate their understanding of their own symptoms before and after the intervention.

**Qualitative data.** Focus group discussions (FGDs) were conducted after the intervention to gain insights into participants' perceptions of the program. These discussions were audio-recorded and transcribed for thematic analysis. Participants were asked to reflect on their knowledge of PH, the usefulness of the I-PSS tool, and any changes in their behavior regarding PH awareness and self-assessment.

## Data analysis

Kolmogorov-Smirnov Test was used for the distribution normality test [21].

**Quantitative data** were analyzed using **paired t-tests** to assess significant differences between pretest and posttest scores for knowledge and I-PSS scores [24–27]. **Cohen's d** was calculated to determine the intervention's effect size on knowledge improvement [18,19]. The **qualitative data** from the focus groups were analyzed thematically using **NVivo software** to identify common themes regarding changes in awareness, the perceived usefulness of the educational materials, and behavioral changes [12–14].

All statistical analyses were conducted using **SPSS version 25**. The level of significance was set at **p < 0.05** for all tests. Steps to calculate effect-size [18,19]:

1. Calculate $SD_{pooled}$:

$$SD_{pooled} = \sqrt{\frac{SD_{pre}^2 + SD_{post}^2}{2}}$$

2. Calculate effect size (Cohen's d):

$$d = \frac{M_{post-test} - M_{pre-test}}{SD_{pooled}}$$

M = Mean pre-test dan post-test.
SD pooled = Combined standard deviation, calculated from the pre-test and post-test SD.

## Ethical considerations

The study was conducted in accordance with ethical guidelines and approved by the Ethical Review Committee of the Faculty of Medicine, Universitas Trisakti under ethical permission number 057/KER/FK/II/2024. The **written informed consent form**, detailing the study's objectives, procedures, potential risks, and benefits, was submitted to the Ethical Review Committee as part of the ethical approval process.

Prior to participation, all participants provided **written informed consent**. The consent process included an explanation of the study's objectives and procedures, and participants were informed that their participation was voluntary and that they could withdraw at any time without consequences. To ensure understanding, participants were asked to read the consent form thoroughly before signing. Any unclear terms were explained verbally by the researchers. The consent forms were signed by participants and witnessed by an impartial third party present during the process. Participants who could not speak Indonesian or were illiterate were excluded from the study. No minors were involved, and thus, parental or guardian consent was not applicable.

To ensure confidentiality, all data were anonymized, and identifying information was not collected. The anonymized datasets were securely stored and accessible only to the research team.

## Results

### 1. Participant characteristics

A total of **32 elderly men** participated in the study. Most participants reported having some prior awareness of urinary symptoms but limited knowledge about **Prostatic Hyperplasia (PH)** and its management options. None of the participants had received prior formal education about PH or the use of the International Prostatic Symptom Score (I-PSS).

The Univariate analysis in this study describes the characteristics of the participants (age), and level of knowledge before (pre-test) and after counseling (post-test).

Before assessing the impact of the educational intervention, it is essential to describe the demographic characteristics of the participants. **Table 2** presents the age distribution of the participants, which may influence their baseline knowledge and learning outcomes.

There were 32 participants out of the 30 planned participants (more than 100% planned).

To determine whether the participant distribution across age groups was signifi -cantly different from an expected equal distribution, a **Chi-Square Goodness of Fit test** was conducted. The results showed no statistically significant difference ($\chi^2(2) = 0.813$, $p = 0.666$), indicating that the age distribution was **homogeneous**. This suggests that age was not a confounding factor in evaluating the impact of the intervention. Further statistical details are provided in Table S3 (Supplementary Materials).

### 2. Quantitative findings

Educational interventions are expected to enhance participants' knowledge, equipping them with essential information about PH symptoms and management. The following analysis presents the impact of the education session.

Table 3 presents the level of knowledge before education. The majority of participants (62.50%) had a moderate level of knowledge, while 28.12% were categorized as having poor knowledge. Only 9.38% demonstrated a good understanding of PH symptoms and management before the session. These findings indicate that prior to the intervention, a significant portion of participants had limited knowledge about the topic, highlighting the need for educational support (as shown in Table 3).

Table 4 illustrates the participants' level of knowledge after the education. A clear improvement was observed in participants' knowledge levels following the educational intervention. The results indicate a positive impact of the intervention,

**Table 2. Age Distribution of Participants.**

| Age (years) | Frequency | % |
|---|---|---|
| 60-65 | 13 | 40.62 |
| 66-70 | 9 | 28.13 |
| >70 | 10 | 31.25 |

**Table 3. Level of knowledge before education.**

| Pre-test Scores | Frequency | % |
|---|---|---|
| Good (85–100) | 3 | 9.38 |
| Moderate (60–80) | 20 | 62.50 |
| Poor (<60) | 9 | 28.12 |

with the proportion of participants in the **good** knowledge category increasing more than doubled, from 9.38% to 21.87%. Notably, no participants were remaining in the **poor** category, suggesting that all individuals improved their understanding. This suggests that even a single educational session can significantly enhance knowledge retention.

Given the importance of structured learning, these findings reinforce the value of community-based education in promoting awareness and proactive health management.

From the comparison of the pre-test results with the post-test, it was found that participants' knowledge and awareness of PH increased after the education was provided.

To ensure the appropriateness of further statistical analyses, a normality test was conducted using the One-Sample Kolmogorov-Smirnov Test. Table 5 presents the results, indicating whether the residual values follow a normal distribution.

The results of the Kolmogorov-Smirnov test show that the significance value is 0.200, which is greater than 0.05. This indicates that the residual values are normally distributed, fulfilling the assumption required for parametric statistical analysis [21].

**Table 4. Level of knowledge after education.**

| Post-test Scores | Frequency | % |
|---|---|---|
| Good (85–100) | 7 | 21.87 |
| Moderate (60–80) | 25 | 78.13 |
| Poor (<60) | 0 | 0 |

**Table 5. One-Sample Kolmogorov-Smirnov Test for Normality Distribution.**

| Output interpretation | One-Sample Kolmogorov-Smirnov Test | Unstandardized Residual |
|---|---|---|
| N | | 32 |
| Normal Parameter[a,b] | Mean | .0000000 |
| | Std. Deviation | 9.86432692 |
| Most Extreme Differences | Absolute | .089 |
| | Positive | .086 |
| | Negative | −0.89 |
| Test Statistic | | 0.89 |
| Asymp. Sig. (2-tailed)[c] | | .200[d] |
| Monte Carlo Sig. (2-tailed)[c] | Sig. | .737 |
| | 99% Confidence Interval Lower Bound | .725 |
| | Upper Bound | .748 |

a. Test distribution is Normal.

b. Calculated from data.

c. Lilliefors Significance Correction.

d. This is a lower bound of the true significance.

e. Lilliefors' method based on 10000 Monte Carlo samples with starting seed 2000000.

Table 6 presents the descriptive statistics for both the pre-test and post-test scores. These statistics provide an overview of the changes in participants' knowledge levels after the educational intervention. By comparing the mean values, standard deviations, and standard error means, we can observe the overall trend of improvement after the intervention.

## Paired samples statistics

This output shows the summary results of descriptive statistics from both pre-test and post-test samples. For the pre-test scores, the average or mean value was 66.25, while the average or mean value of the post-test results was 75.1563. The standard deviation value for the pre-test is 11.57026 and the post-test is 10.03899. The standard Error Mean of the pre-test is 2.04535 and the post-test is 1.77466. Because the average post-test score of 75.1563 is greater than the pre-test score of 66.25, descriptively there is an increase in the average post-test score compared to the pre-test score [24–27]. Table 7 presents the correlation analysis between pre-test and post-test scores. This analysis examines whether there is a significant relationship between participants' scores before and after the intervention. The correlation coefficient and significance values help determine the strength and direction of this association.

The output in the second section is the outcome of the relationship or correlation between the two variables or data, specifically the pre-test and post-test. The output above shows the results of the correlation test or the relationship between the pre- and post-test scores. Based on the output above, it is known that the correlation coefficient value is 0.186 with a significance value of 0.309. Because the significance value is 0.309 > 0.05, it is possible to draw the conclusion that there is no meaningful relationship between the pre-test and post-test variables [24–27].

Given that there was no significant correlation, a paired samples t-test was conducted to examine whether there was a statistically significant difference between pre-test and post-test scores [24–27].

**Table 6. Sample Paired T-Test Output interpretation.**

| Sample Paired T-Test | | | | | |
|---|---|---|---|---|---|
| **Output interpretation** | | | | | |
| | | **Paired Samples Statistics** | | | |
| | | **Mean** | **N** | **Std. Deviation** | **Std. Error Mean** |
| Pair 1 | Pre Test | 66.2500 | 32 | 11.57026 | 2.04535 |
| | Post Test | 75.1563 | 32 | 10.03899 | 1.77466 |

**Table 7. Pair Samples Correlations.**

| | | **Paired Samples Correlations** | | | |
|---|---|---|---|---|---|
| | | | | | **Significance** |
| | | **N** | **Correlations** | **One-sided p** | **Two-sided p** |
| **Pair 1** | **Pre test & Post test** | 32 | .186 | .154 | .309 |

**Table 8. Paired Samples Test.**

| | | **Paired Samples Test** | | | | | | | | |
|---|---|---|---|---|---|---|---|---|---|---|
| | | **Paired Differences** | | | | | | | **Significance** | |
| | | | | | **95% Confidence Interval of the Difference** | | | | | |
| | | **Mean** | **Std. Deviation** | **Std. Error Mean** | **Lower** | **Upper** | **t** | **df** | **One-Sided p** | **Two-Sided p** |
| Pair 1 | Pre test-Post test | −8.90625 | 13.83861 | 2.44634 | −13.89560 | −3.91690 | −3.641 | 31 | <,001 | <,001 |

Table 8 presents the results of the paired samples t-test, which was conducted to determine whether there is a statistically significant difference between the pre-test and post-test scores. This test assesses whether the observed changes in participants' performance after the intervention are likely due to the treatment rather than random variation. By analyzing the mean difference, confidence intervals, and significance values, we can evaluate the effectiveness of the intervention. Furthermore, the paired samples t-test was performed to assess whether the observed difference in pre-test and post-test scores is statistically significant. The results show that the mean difference between the two tests is 8.90625, with a standard deviation of 13.83861. The two-sided significance value of <0.001 is smaller than 0.05, indicating that Ha is accepted. Therefore, it can be concluded that there is a significant difference between the pre-test and post-test results. This finding confirms that the intervention or treatment applied between the two tests had a measurable impact on participants' performance [24–27].

After the paired sample t-test showed a significant difference (p<0.001), we calculated the effect size using Cohen's d to assess the strength of the intervention impact.

Steps to calculate effect-size

1. Calculate **SD** $_{pooled}$:

$$SD_{pooled} = \sqrt{\frac{(11,570262^2 + 10,038992^2)}{2}}$$

$$\approx 10,955$$

2. Calculate **Cohen's d**:

$$d = \frac{75,1563 - 66,25}{10,955} \approx 0,82$$

Thus, the effect size (Cohen's d) = 0.82, which indicates a large impact of the educational intervention [14,15]. Paired-sample t-test showed a significant difference between pre-test and post-test knowledge scores (pre-test average = 66.25, post-test average = 75.16, p < 0.001). The effect size value (Cohen's d) of 0.82 shows that education has a significant impact (classified as large) and is practically relevant in increasing participants' knowledge [18,19].

3. Qualitative Findings

The qualitative findings supported and enriched these results. Thematic analysis of focus group discussions revealed three major themes reflecting participants' experiences and perceptions of the intervention:

**1. Increased understanding of PH symptoms and management**

Participants reported a clearer understanding of PH symptoms, such as urinary frequency and nocturia, and their association with aging. Many expressed confidence in identifying these symptoms early and seeking medical advice.

**2. Empowerment through the I-PSS tool**

The introduction of the I-PSS form was highly appreciated, as participants found it a practical and easy-to-use tool for self-assessment. Several participants expressed that using the tool made them feel more in control of their health.

### 3. Suggestions for program improvement

While participants valued the session, some suggested that more interactive activities, such as role-playing or case studies, could enhance their understanding. A few also recommended involving family members in future sessions to raise broader awareness.

### 4. Overall impact of the intervention

The combined findings from quantitative and qualitative analyses highlight the effectiveness of the educational program in improving participants' knowledge and awareness of PH. The significant changes in pretest and posttest scores, coupled with positive feedback from participants, suggest that community-based interventions using tools like I-PSS can have a meaningful impact on promoting early detection and proactive management of PH among elderly populations.

## Discussion

This study evaluated the effectiveness of a community-based educational intervention in enhancing knowledge and awareness about **Prostatic Hyperplasia (PH)** among elderly men in Bogor, Indonesia. The findings demonstrate significant improvements in participants' knowledge and awareness of PH, as evidenced by increases in knowledge scores and self-reported understanding of symptoms using the **International Prostatic Symptom Score (I-PSS)**. The results highlight the potential of structured community-based programs to address gaps in health education for elderly populations.

### 1. Interpretation of findings

The significant increase in knowledge scores from pretest to posttest underscores the effectiveness of the intervention. The mean pretest score of **66.25 ± 11.57** increased to **75.16 ± 10.04**, reflecting a large effect size (Cohen's d = 0.82). These results align with previous studies indicating that targeted health education can substantially improve knowledge and promote health-seeking behaviors in underserved populations.

The improvement in **I-PSS scores** further suggests that participants gained a better understanding of their urinary symptoms, potentially enabling them to seek medical care earlier [25–27]. Notably, the relatively uniform education level among participants may have contributed to minimizing variability in knowledge acquisition, allowing for a more consistent intervention effect. This uniformity serves as a strength of the study, as it reduces potential confounding effects related to baseline disparities in health literacy.

Qualitative findings provided deeper insights into participants' experiences.

Participants appreciated the use of the I-PSS tool, reporting that it empowered them to self-assess their symptoms and monitor their health. This suggests that integrating practical tools like the I-PSS into community health programs can enhance their impact by promoting autonomy in managing chronic conditions. However, some participants expressed a desire for more interactive components, such as role-playing or family involvement, to reinforce learning and encourage broader awareness [27–29].

### 2. Novelty and contribution to the literature

This study is among the first to introduce the **I-PSS tool** in a non-clinical, community-based educational setting, highlighting its feasibility and effectiveness as a public health education tool. Previous studies on PH education have predominantly been conducted in clinical contexts, focusing on patients already seeking care. By contrast, this study targeted individuals within the community, enabling broader access to education and promoting proactive health management. The findings also address a critical research gap regarding the use of structured educational interventions for elderly populations in low-resource settings. The single-session format demonstrated that even brief interventions can yield significant improvements in knowledge and awareness, making it a scalable and cost-effective approach for similar populations worldwide.

The findings also contribute to the growing body of evidence supporting the integration of symptom assessment tools into community-based health education programs. The successful implementation of the I-PSS in this setting suggests that similar tools could be adapted for other chronic conditions to facilitate early detection and encourage timely medical consultation.

### 3. Implications for public health

The implications of this study extend beyond the immediate community. The scalable nature of the intervention suggests its applicability to other low- and middle-income countries (LMICs) facing similar challenges with PH awareness and healthcare access. By empowering individuals to self-assess and seek timely medical advice, programs like this can reduce the burden of late-stage diagnoses and associated complications, ultimately alleviating healthcare costs. Moreover, the inclusion of practical tools like the I-PSS provides a framework for integrating patient-centered approaches into community health initiatives. These findings support the integration of health education into routine public health campaigns, particularly in regions with limited access to healthcare infrastructure.

### 4. Limitations and directions for future research

Despite its promising findings, this study has several limitations. First, the quasi experimental design lacked a control group, which limits the ability to attribute observed improvements solely to the intervention. Future studies should consider including control groups to strengthen causal inferences. Second, the sample size was relatively small, and the study was conducted in a single community, which may limit the generalizability of the findings. Expanding the study to include diverse settings and larger populations would provide more robust evidence of the intervention's effectiveness. Additionally, while the I-PSS tool was well-received, future research could explore the long-term impact of such interventions on health outcomes, including symptom management and healthcare utilization. Further studies could also assess the effectiveness of incorporating family members into the educational process to enhance support systems for elderly individuals.

## Conclusions

This study demonstrates the effectiveness of a single-session, community-based educational intervention in improving knowledge and awareness about **Prostatic Hyperplasia (PH)** among elderly men. By integrating the **International Prostatic Symptom Score (I-PSS)** as a public health education tool, this study highlights a novel approach to addressing health education gaps in low-resource settings.

The significant improvements in knowledge and awareness observed in this study underscore the potential of such interventions to promote early detection and proactive management of PH. These findings have broader implications for public health strategies, suggesting that scalable, community-centered approaches can contribute to reducing the healthcare burden associated with late-stage PH diagnoses. However, due to the relatively homogeneous educational background of participants, the influence of education level on the effectiveness of the intervention could not be analyzed. Therefore, our findings focus on the overall knowledge improvement observed before and after the educational intervention without comparing its impact based on education level.

These findings have broader implications for public health strategies, suggesting that scalable, community-centered approaches can contribute to reducing the healthcare burden associated with late-stage PH diagnoses.

Future research should aim to validate these findings in diverse settings, incorporate control groups, and evaluate long-term impacts on health behaviors and outcomes. Expanding the scope of such programs to include family involvement and interactive methods could further enhance their effectiveness and reach.

## Supporting information

**S1 File. Supplementary Materials. This document contains additional data that support the findings presented in the manuscript.**
(PDF)

## Author contributions

**Conceptualization:** Hari Krismanuel.

**Data curation:** Purnamawati Tjhin.

**Formal analysis:** Hari Krismanuel.

**Investigation:** Hari Krismanuel.

**Methodology:** Hari Krismanuel.

**Visualization:** Purnamawati Tjhin.

**Writing – original draft:** Hari Krismanuel.

**Writing – review & editing:** Hari Krismanuel.

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
