## [Decision Letter · Decision Letter 0]

6 Feb 2025

Dear Dr. Krismanuel,

If applicable, we recommend that you deposit your laboratory protocols in protocols.io to enhance the reproducibility of your results. Protocols.io assigns your protocol its own identifier (DOI) so that it can be cited independently in the future. For instructions see: https://journals.plos.org/plosone/s/submission-guidelines#loc-laboratory-protocols . Additionally, PLOS ONE offers an option for publishing peer-reviewed Lab Protocol articles, which describe protocols hosted on protocols.io. Read more information on sharing protocols at https://plos.org/protocols?utm_medium=editorial-email&utm_source=authorletters&utm_campaign=protocols.

We look forward to receiving your revised manuscript.

Kind regards,

Mukhtiar Baig, Ph.D.

Academic Editor

PLOS ONE

Journal requirements:   When submitting your revision, we need you to address these additional requirements. 1. Please ensure that your manuscript meets PLOS ONE's style requirements, including those for file naming. The PLOS ONE style templates can be found at https://journals.plos.org/plosone/s/file?id=wjVg/PLOSOne_formatting_sample_main_body.pdf and https://journals.plos.org/plosone/s/file?id=ba62/PLOSOne_formatting_sample_title_authors_affiliations.pdf. 2. Please amend either the title on the online submission form (via Edit Submission) or the title in the manuscript so that they are identical. 3. Please match your authorship list in your manuscript file and in the system. 4. We note that you have indicated that there are restrictions to data sharing for this study. For studies involving human research participant data or other sensitive data, we encourage authors to share de-identified or anonymized data. However, when data cannot be publicly shared for ethical reasons, we allow authors to make their data sets available upon request. For information on unacceptable data access restrictions, please see http://journals.plos.org/plosone/s/data-availability#loc-unacceptable-data-access-restrictions.  Before we proceed with your manuscript, please address the following prompts: a) If there are ethical or legal restrictions on sharing a de-identified data set, please explain them in detail (e.g., data contain potentially identifying or sensitive patient information, data are owned by a third-party organization, etc.) and who has imposed them (e.g., a Research Ethics Committee or Institutional Review Board, etc.). Please also provide contact information for a data access committee, ethics committee, or other institutional body to which data requests may be sent. b) If there are no restrictions, please upload the minimal anonymized data set necessary to replicate your study findings to a stable, public repository and provide us with the relevant URLs, DOIs, or accession numbers. Please see http://www.bmj.com/content/340/bmj.c181.long for guidelines on how to de-identify and prepare clinical data for publication. For a list of recommended repositories, please see https://journals.plos.org/plosone/s/recommended-repositories. You also have the option of uploading the data as Supporting Information files, but we would recommend depositing data directly to a data repository if possible. Please update your Data Availability statement in the submission form accordingly.

Reviewers' comments:

Reviewer's Responses to Questions

**Comments to the Author**

1. Is the manuscript technically sound, and do the data support the conclusions?

Reviewer #1: Yes

Reviewer #2: Partly

Reviewer #3: Yes

Reviewer #4: Yes

2. Has the statistical analysis been performed appropriately and rigorously?

Reviewer #1: Yes

Reviewer #2: Yes

Reviewer #3: Yes

Reviewer #4: Yes

3. Have the authors made all data underlying the findings in their manuscript fully available?

Reviewer #1: Yes

Reviewer #2: No

Reviewer #3: Yes

Reviewer #4: Yes

4. Is the manuscript presented in an intelligible fashion and written in standard English?

Reviewer #1: Yes

Reviewer #2: Yes

Reviewer #3: Yes

Reviewer #4: Yes

Reviewer #1: The manuscript presents a community-based educational intervention aimed at improving knowledge and awareness about Prostatic Hyperplasia (PH) among elderly men in Bogor, Indonesia. Overall, the study contributes meaningfully to the field, especially in addressing health education gaps for elderly populations in low-resource settings. Below is the detailed feedback regarding the manuscript:

Technical Soundness and Data Support: The manuscript describes a well-structured quasi-experimental study with a pretest-posttest design. The statistical analysis, including paired t-tests and Cohen’s d calculation, is appropriately applied to assess the intervention's effectiveness. The quantitative results are compelling, with a significant increase in knowledge scores post-intervention. Qualitative data enrich the findings by providing deeper insights into participant perceptions. However, the lack of a control group limits the ability to establish causality. Future iterations could consider incorporating control groups to strengthen the validity of the conclusions.

Statistical Analysis: The statistical methods employed, including normality tests and effect size calculations, are rigorous and align with the study's objectives. The authors have adequately described the steps taken to ensure the robustness of the analysis. The effect size (Cohen’s d = 0.82) indicates a large practical impact of the intervention, which is encouraging.

Data Availability: The data availability statement is adequate, and all relevant data are included within the manuscript and its supporting files. However, it would be beneficial for the authors to specify whether the raw dataset (e.g., anonymized pretest and posttest scores) is available in a public repository for reproducibility.

Language and Presentation: The manuscript is written in clear and standard English, making it accessible to a wide audience. The structure of the paper is logical, and the arguments are easy to follow. While there are no major grammatical errors, minor typographical errors should be addressed during revision.

Strengths of the Study: The integration of the International Prostatic Symptom Score (I-PSS) into a community education setting is novel and provides a practical tool for participants to self-assess their symptoms. The mixed-methods approach adds depth to the findings by combining quantitative results with qualitative insights. The scalability and low-cost nature of the intervention make it suitable for broader applications in similar settings.

Limitations and Suggestions for Improvement: The lack of a control group is a significant limitation. Future studies should aim to include a control group to strengthen causal inferences. The sample size, while adequate for initial findings, could be expanded to improve generalizability. Including family members in the educational sessions may enhance the program’s impact and encourage broader awareness. More interactive elements, such as case studies or role-playing, could further engage participants and reinforce learning.

Ethics and Reporting Standards: The study adheres to ethical standards, with appropriate approval obtained and clear documentation of informed consent procedures. The manuscript follows reporting guidelines and includes sufficient methodological details to ensure reproducibility.

Conclusion: The study provides strong evidence for the effectiveness of community-based education using the I-PSS tool. It demonstrates potential as a scalable, low-cost intervention to address health education gaps in low-resource settings. The authors have made a valuable contribution to the field of community health education.

Reviewer #2: The article is an interesting one but lacks few basic components, like the purpose of study is not clear. The article is about educational intervention and patient education but maximum emphasis is on the statistical details. The pre-test/post-test questionnaire is not provided nor discussed. The results just mention the difference between cumulative score without details of components (like symptoms, management options etc).

Statistical details may be reviewed by a stastitician

Reviewer #3: The data regarding the education level of participants should be added to this study to identify the relation between the education level and the knowledge of prostate hyperplasia. Therefore, the conclusion should also mention the effectiveness of education level and the impact of health education in the community regarding prostate hyperplasia

Reviewer #4: This studies can be applied in daily urology clinical setting. the data used in this research is reliable and has been handled appropriately. The next research about early screening and treatment of benign prostatic hyperplasia can be developed from this studies, by taking larger sampels or populations

**Do you want your identity to be public for this peer review?** For information about this choice, including consent withdrawal, please see our Privacy Policy

Reviewer #1: **Yes: ** Dr.dr.Reza Aditya Digambiro, M.Kes, M.Ked(PA), Sp.PA

Reviewer #2: No

Reviewer #3: No

Reviewer #4: No

---

## [Author Response · Author response to Decision Letter 1]

20 Mar 2025

To: em@editorialmanager.com

Subject: Response to Minor Revision – Manuscript ID: PONE-D-24-53996

Dear Mukhtiar Baig, Ph.D.,

Thank you for your consideration of our manuscript, "Examining the Effectiveness of Prostatic Hyperplasia Education on the Level of Participants' knowledge and awareness." We appreciate the valuable feedback from editor and the reviewers and have carefully addressed all their comments in the revised manuscript.

In particular, we have responded to concerns regarding participant data, statistical focus, and clinical implications in our detailed responses to the reviewers' questions and requests. Additionally, we have revised the manuscript to enhance clarity and ensure a balanced discussion between statistical findings and clinical relevance.

As per the request for data, we have included the datasets related to management options for PH, anonymized participant data, output of data analysis, and pre- and post-test results in the Supplementary Material. Furthermore, the data has been deposited in Figshare following the editorial recommendation.

We have attached the revised manuscript along with a point-by-point response. Please let us know if any further modifications are needed.

Thank you for your time and consideration. We look forward to your feedback.

Best regards,

[Dr. Hari Krismanuel]

[Universitas Trisakti]

[hari_krismanuel@trisakti.ac.id]

RESPONSE TO REVIEWERS

1. Is the manuscript technically sound, and does the data support the conclusions?

Reviewer #1: Yes

Reviewer #2: Partly

Reviewer #3: Yes

Reviewer #4: Yes

RESPONSE TO REVIEWERS

Dear Reviewers,

We sincerely appreciate your thorough review of our manuscript titled "Examining the Effectiveness of Prostatic Hyperplasia Education on the Level of Participants' Knowledge and Awareness." We are grateful for your constructive feedback, which has helped us refine and improve the clarity and rigor of our study.

Regarding the question posed by the editor on whether the manuscript is technically sound and whether the data support our conclusions, we note that three reviewers (Reviewers #1, #3, and #4) responded affirmatively, while Reviewer #2 indicated "Partly." However, no specific concerns were provided regarding which aspect of the study was deemed partial in technical soundness.

To address the concerns raised by Reviewer #2, we have made the following clarifications and improvements:

Response to Reviewer #2:

Thank you for your valuable feedback. We appreciate your insights and have carefully addressed your concerns regarding the technical soundness of the manuscript and the adequacy of the data in supporting our conclusions.

1. Study Design and Rationale

Our study employs a well-established quasi-experimental pretest-posttest design without a control group, which is appropriate for evaluating educational interventions. We are confident that our study meets the technical rigor required for evaluating educational interventions, and that the data presented provide strong support for our conclusions. While a control group could have provided additional comparison, our focus was to assess within-group knowledge improvement directly attributable to the intervention.

2. Statistical Analysis and Transparency

The statistical analysis was conducted independently by the authors using SPSS, applying appropriate methods such as paired t-tests and effect size calculations (Cohen’s d) to comprehensively assess the intervention’s impact. We explicitly described this in the Methods section and presented the detailed statistical results in the Results section to ensure full transparency.

3. Clarification on the Pretest-Posttest Questionnaire

The questionnaire was adapted from the validated International Prostate Symptom Score (I-PSS) tool. We have now included a clearer explanation of its components and the rationale for its use in the revised manuscript.

4. Balance Between Statistical analysis, Quantitative and Qualitative Findings

While statistical analysis was essential in demonstrating the intervention’s effectiveness, we also presented qualitative findings that provided deeper insights into participants' understanding and perceptions. These qualitative results are now emphasized in both the Results and Discussion sections to ensure a balanced perspective.

We believe these revisions further strengthen the manuscript's clarity and rigor. If Reviewer #2 has specific concerns beyond these points, we would greatly appreciate further clarification.

Best regards,

[Dr. Hari Krismanuel]

[Universitas Trisakti]

2. Has the statistical analysis been performed appropriately and rigorously?

Reviewer #1: Yes

Reviewer #2: Yes

Reviewer #3: Yes

Reviewer #4: Yes

RESPONSE TO REVIEWERS

Dear Reviewers,

We sincerely appreciate the reviewers’ positive evaluations of our statistical analysis. As all four reviewers (Reviewers #1, #2, #3, and #4) have confirmed that the statistical analysis was performed appropriately and rigorously, we have maintained our analytical approach in the revised manuscript.

Nonetheless, we have carefully reviewed the statistical methods to ensure clarity and have provided additional explanations where necessary to enhance the transparency of our analysis. We appreciate the reviewers’ recognition of the robustness of our statistical approach and thank them for their valuable insights.

Best regards,

[Dr. Hari Krismanuel]

[Universitas Trisakti]

3. Have the authors made all data underlying the findings in their manuscript fully available?

The PLOS Data policy requires authors to make all data underlying the findings described in their manuscript fully available without restriction, with rare exceptions (please refer to the Data Availability Statement in the manuscript PDF file). The data should be provided as part of the manuscript or its supporting information, or deposited to a public repository. For example, in addition to summary statistics, the data points behind means, medians, and variance measures should be available. If there are restrictions on publicly sharing data—e.g. participant privacy or use of data from a third party—those must be specified.

Reviewer #1: Yes

Reviewer #2: No

Reviewer #3: Yes

Reviewer #4: Yes

RESPONSE TO REVIEWERS

Dear Reviewers,

We sincerely appreciate your comments regarding data availability. While three reviewers (Reviewers #1, #3, and #4) confirmed that the data have been made fully available, we acknowledge the concern raised by Reviewer #2.

To clarify, we fully support data transparency while adhering to ethical guidelines and participant privacy protection. The original dataset contains sensitive personal information, including participant names and addresses, which must remain confidential. However, in response to these concerns, we have taken the following steps:

• Request for Clarification: We respectfully request Reviewer 2 to provide further clarification regarding the reasons behind their "No" response. This will enable us to better address their specific concerns and ensure that we have provided all necessary data in an appropriate format.

• Anonymization of data: We have prepared a version of the dataset where participant names are represented only by initials, and all address information has been removed.

• We have added the Data Availability Statement section to explicitly mention the anonymization process and to clarify that anonymized participant data are securely stored and available as Supplemental Materials through [repository link]."

• To further enhance transparency and address the reviewer's concern about the availability of the questionnaire and detailed results, we will include the complete pre-test and post-test questionnaire, the detailed SPSS output of our statistical analysis, and a brief summary of the management options discussed during the educational session as Supplemental Materials. This will provide a more comprehensive understanding of the assessment tools and the detailed findings of our study.

We believe these measures address Reviewer #2’s concerns while maintaining the integrity and transparency of our research.

Best regards,

[Dr. Hari Krismanuel]

[Universitas Trisakti]

4. Is the manuscript presented in an intelligible fashion and written in standard English?

Reviewer #1: Yes

Reviewer #2: Yes

Reviewer #3: Yes

Reviewer #4: Yes

RESPONSE TO REVIEWERS

Dear Reviewers,

We sincerely appreciate your feedback on the clarity and readability of our manuscript. We are pleased to note that all reviewers have confirmed that the manuscript is presented in an intelligible fashion and written in standard English.

Thank you for your time and valuable insights.

Best regards,

[Dr. Hari Krismanuel]

[Universitas Trisakti]

5. Review Comments to the Author

Reviewer #1: The manuscript presents a community-based educational intervention aimed at improving knowledge and awareness about Prostatic Hyperplasia (PH) among elderly men in Bogor, Indonesia. Overall, the study contributes meaningfully to the field, especially in addressing health education gaps for elderly populations in low-resource settings. Below is the detailed feedback regarding the manuscript:

Technical Soundness and Data Support: The manuscript describes a well-structured quasi-experimental study with a pretest-posttest design. The statistical analysis, including paired t-tests and Cohen’s d calculation, is appropriately applied to assess the intervention's effectiveness. The quantitative results are compelling, with a significant increase in knowledge scores post-intervention. Qualitative data enrich the findings by providing deeper insights into participant perceptions. However, the lack of a control group limits the ability to establish causality. Future iterations could consider incorporating control groups to strengthen the validity of the conclusions.

Statistical Analysis: The statistical methods employed, including normality tests and effect size calculations, are rigorous and align with the study's objectives. The authors have adequately described the steps taken to ensure the robustness of the analysis. The effect size (Cohen’s d = 0.82) indicates a large practical impact of the intervention, which is encouraging.

Data Availability: The data availability statement is adequate, and all relevant data are included within the manuscript and its supporting files. However, it would be beneficial for the authors to specify whether the raw dataset (e.g., anonymized pretest and posttest scores) is available in a public repository for reproducibility.

Language and Presentation: The manuscript is written in clear and standard English, making it accessible to a wide audience. The structure of the paper is logical, and the arguments are easy to follow. While there are no major grammatical errors, minor typographical errors should be addressed during revision.

Strengths of the Study: The integration of the International Prostatic Symptom Score (I-PSS) into a community education setting is novel and provides a practical tool for participants to self-assess their symptoms. The mixed-methods approach adds depth to the findings by combining quantitative results with qualitative insights. The scalability and low-cost nature of the intervention make it suitable for broader applications in similar settings.

Limitations and Suggestions for Improvement: The lack of a control group is a significant limitation. Future studies should aim to include a control group to strengthen causal inferences. The sample size, while adequate for initial findings, could be expanded to improve generalizability. Including family members in the educational sessions may enhance the program’s impact and encourage broader awareness. More interactive elements, such as case studies or role-playing, could further engage participants and reinforce learning.

Ethics and Reporting Standards: The study adheres to ethical standards, with appropriate approval obtained and clear documentation of informed consent procedures. The manuscript follows reporting guidelines and includes sufficient methodological details to ensure reproducibility.

Conclusion: The study provides strong evidence for the effectiveness of community-based education using the I-PSS tool. It demonstrates potential as a scalable, low-cost intervention to address health education gaps in low-resource settings. The authors have made a valuable contribution to the field of community health education.

Reviewer #2: The article is an interesting one but lacks a few basic components, like the purpose of the study is not clear. The article is about educational intervention and patient education but the maximum emphasis is on the statistical details. The pre-test/post-test questionnaire is not provided nor discussed. The results just mention the difference between cumulative scores without details of components (like symptoms, management options, etc).

Statistical details may be reviewed by a statistician

Reviewer #3: The data regarding the education level of participants should be added to this study to identify the relation between the education level and the knowledge of prostate hyperplasia. Therefore, the conclusion should also mention the effectiveness of education level and the impact of health education in the community regarding prostate hyperplasia

Reviewer #4: This study can be applied in a daily urology clinical setting. The data used in this research is reliable and has been handled appropriately. The next research about early screening and treatment of benign prostatic hyperplasia can be developed from these studies, by taking larger samples or populations.

Response to Reviewers

Dear Reviewers,

We sincerely appreciate the constructive feedback on our manuscript. Below, we address each comment in detail and describe the revisions we have made to enhance the clarity and rigor of our study.

RESPONSE TO REVIEWER #1

Dear Reviewer #1,

We sincerely appreciate your thorough and constructive review of our manuscript. Your comments have been very helpful in improving the clarity and rigor of our study. Below are our detailed responses to your feedback:

1. Technical Soundness and Data Support

Thank you for acknowledging the structured approach of our quasi-experimental study and the robustness of our statistical analysis. Regarding your concern about the lack of a control group, we acknowledge that having a control group would strengthen causal inferences. However, our study was designed as a one-group pretest-posttest study due to practical and ethical considerations. Since the primary goal of this research was to evaluate the effectiveness of an educational intervention in improving knowledge and awareness of Prostatic Hyperplasia (PH) among elderly men, we focused on measuring individual changes before and after the intervention within the same group of participants. This approach allowed us to assess the direct impact of the education without withholding valuable health information from any participants, which would have been required in a controlled setting.

Additionally, the statistical significance of our findings (Cohen’s d = 0.82) indicates a strong effect size, supporting the effectiveness of the intervention. While a future study incorporating a control group would be ideal, we believe that our current approach still provides valuable insights, particularly in low-resource settings where access to healthcare educati

---

## [Editor Report · Decision Letter 1]

18 May 2025

Examining the effectiveness of prostatic hyperplasia education on the level of participant’s knowledge and awareness

PONE-D-24-53996R1

Dear Dr. Krismanuel,

We’re pleased to inform you that your manuscript has been judged scientifically suitable for publication and will be formally accepted for publication once it meets all outstanding technical requirements.

Kind regards,

Mukhtiar Baig, Ph.D.

Academic Editor

PLOS ONE

---

## [Editor Report · Acceptance letter]

PONE-D-24-53996R1

PLOS ONE

Dear Dr. Krismanuel,

I'm pleased to inform you that your manuscript has been deemed suitable for publication in PLOS ONE. Congratulations! Your manuscript is now being handed over to our production team.

Kind regards,

on behalf of

Professor Mukhtiar Baig

Academic Editor

PLOS ONE